# The Automatic Solution of Macromolecular Crystal Structures via Molecular Replacement Techniques: REMO22 and Its Pipeline

**DOI:** 10.3390/ijms24076070

**Published:** 2023-03-23

**Authors:** Benedetta Carrozzini, Giovanni Luca Cascarano, Carmelo Giacovazzo

**Affiliations:** Istituto di Cristallografia, The National Research Council (CNR), Via G. Amendola 122/o, I-70126 Bari, Italy; benedetta.carrozzini@ic.cnr.it (B.C.); gianluca.cascarano@ic.cnr.it (G.L.C.)

**Keywords:** molecular replacement, proteins, nucleic acids, automated pipeline

## Abstract

A description of REMO22, a new molecular replacement program for proteins and nucleic acids, is provided. This program, as with REMO09, can use various types of prior information through appropriate conditional distribution functions. Its efficacy in model searching has been validated through several test cases involving proteins and nucleic acids. Although REMO22 can be configured with different protocols according to user directives, it has been developed primarily as an automated tool for determining the crystal structures of macromolecules. To evaluate REMO22’s utility in the current crystallographic environment, its experimental results must be compared favorably with those of the most widely used Molecular Replacement (MR) programs. To accomplish this, we chose two leading tools in the field, PHASER and MOLREP. REMO22, along with MOLREP and PHASER, were included in pipelines that contain two additional steps: phase refinement (SYNERGY) and automated model building (CAB). To evaluate the effectiveness of REMO22, SYNERGY and CAB, we conducted experimental tests on numerous macromolecular structures. The results indicate that REMO22, along with its pipeline REMO22 + SYNERGY + CAB, presents a viable alternative to currently used phasing tools.

## 1. Introduction

The practical solution to the phase problem for small to medium-sized molecules containing up to 300 non-H atoms in the asymmetric unit has been achieved. Several well-documented computer programs that represent this accomplishment include SnB [1,2] SHELX-D [3], ACORN [4], SUPERFLIP [5], SIR2002 [6], SIR2004 [7] (the acronym SIR is associated with *semi-invariant representations* [8,9], which is a general theory that explains the role of structure invariants and semi-invariants in the phasing process using Direct Methods).

Ab initio techniques in the macromolecular field have not achieved the same level of success. To succeed, at least one of the following two strict conditions must be met: atomic or quasi-atomic data resolution for Direct Methods, or the presence of heavy atoms in the unit cell for Patterson Deconvolution Techniques. The largest unknown protein that was solved ab initio by Direct Methods prior to 2006 was cytochrome C3 (PDB code: 1gyo; [10]), with 2003 non-H atoms in the asu, solved by SHELX-D. In 2006, Mooers & Matthews [11] solved the unknown structure of the bacteriophage P22 lysozyme (PDB code: 2anv), which has 2268 non-H atoms in the asymmetric unit, using SIR2002. Patterson deconvolution techniques [12,13,14,15,16,17] were able to solve large-sized protein structures at non-atomic resolution (e.g., [18], 1e3u, with about 7890 non-H atoms in the asymmetric unit and 1.65 Å of data resolution), and also achieved success with 1buu, a protein with 1283 non-H atoms in the asymmetric unit and 1.92 Å data resolution.

Among the non-ab initio techniques for solving the phase problem in macromolecular crystallography, MR has been the most successful so far, with a higher probability of automatically determining macromolecular structures [19,20,21]. Researchers have attempted to search in the six-dimensional space, with efforts by Kissinger et al. [22], Jamrog et al. [23], Glykos & Kokkinidis [24], Fujinaga & Read [25], among others. However, some authors have preferred to split the expensive six-dimensional search into two steps, the rotation and translation step, as done by AMoRe [26], BEAST [27], MOLREP [28], PHASER [29], REMO09 [30], and ARCIMBOLDO [31,32,33].

This paper will only focus on MR approaches to the phase problem. The full automation of the crystal structure solution via MR requires four steps to be completed successfully:(i)Finding a good enough model. If the sequence identity between the known structure and target is low or limited, the solution of the phase problem may be hindered.(ii)An efficient MR program to orient and translate the model molecules correctly into the target asymmetric unit. This program must be able to handle cases where the model search is far from optimal, as even well-defined rotation and translation parameters can lead to a large mean phase error.(iii)The phase extension and refinement step. The MR modulus often produces a large phase error on a limited number of reflections. This step is usually accomplished through electron density modification (EDM) techniques included in large crystallographic packages such as CNS [34], CCP4 [35], SHARP [36], PHENIX [37] and the SHELX series [38]. Burla et al. [39] described a procedure, named SYNERGY, which combines DM by Cowtan [40] with out-of-mainstream techniques such as *free lunch* [41,42], low-density Fourier transform [43], *vive la difference* [44,45], phantom derivative [46,47], and phase driven model refinement [48].(iv)An automated model building (AMB) program to generate a model that fits the experimental data. Popular AMB programs include BUCCANEER [49] for proteins, NAUTILUS [50] for nucleic acids, ARP/wARP [51] for proteins and nucleic acids, and the PHENIX AUTOBUILD wizard [52] for proteins and nucleic acids. Recently, a new cyclic AMB procedure called CAB [53], which uses BUCCANEER for protein model building and NAUTILUS for nucleic acids building, has been developed and shown to be highly efficient in experimental applications (see Papers I–III [54,55,56]).

Improving step (i) can greatly enhance the success and automation of MR phasing processes. The success rate of MR is primarily dependent on the root-mean-squared distance between the atomic positions of the template and the target structure. As sequence identity (SI) between the target and the model decreases, the success rate usually increases (although there are no universal cutoffs for SI, SI < 0.30 is generally considered the lower limit for MR success). Despite the increasing number of structures providing good coverage of protein families, the difficulties in this area arise from the non-negligible percentage of proteins sequences without structural homologues. To tackle this issue, automated pipelines have been developed to discover and prepare numerous search models, which can then be processed by MR programs. For instance, for the 1w2y structure, the pipeline MrBUMP by Keegan & Winn [57] selected five models with varying degrees of usefulness.

Point (i) is beyond the scope of this paper, as we will focus solely on steps (ii), (iii) and (iv). Nevertheless, we acknowledge that integrating our techniques with homology detection programs could enhance their potential further. Additionally, the relationship between MR techniques and advanced machine-learning-based structure prediction algorithms, such as AlphaFold [58], will not be covered in this paper. While these methods can predict substantial regions of a protein structure accurately based on its amino acid sequence, experimental data must verify such predictions. MR procedures may play a central role in this area, and a two-way relationship is expected. For instance, the information contained in an MR density map may improve the accuracy of the AlphaFold modelling [59]. Likewise, the predicted models could facilitate a more straightforward application of MR techniques.

This is the fourth paper in a series dedicated to the automatic crystal structure solution of macromolecular structures. Our goal is to achieve, as in the case of small molecules, a high percentage of practical MR cases solved automatically through a pipeline that requires minimal input, allowing users more time to focus on final model refinement. However, it is important to note that REMO22 is not a completely directive-free program, as directives may be necessary to change, e.g., the MR model or define the estimated number of model copies in the target asymmetric unit. To understand how this paper builds on previous work, we need to revisit Papers I–III. Paper I demonstrated the effectiveness of phase refinement by SYNERGY and the high quality of the CAB automated model building for proteins, while revealing the inadequacy of REMO09 and AMB programs for nucleic acid structures. Papers II and III were dedicated to extending CAB to nucleic acids, and now we return to the MR step to present REMO22, which is an effective successor to REMO09.

To evaluate whether REMO22 can truly be a viable alternative to the most used MR programs, we compared its performance to that of MOLREP (version 11.7.03) and PHASER (version 2.8.3) using the same set of test structures. We chose these programs because they have been used to solve a high percentage of published structures (approximately 61,000 solved by PHASER and 24,000 solved by MOLREP out of over 200,000 structures in the PDB). Additionally, the two programs use different theoretical approaches: PHASER operates in reciprocal space and relies heavily on maximum likelihood techniques, while MOLREP orients model copies using the Patterson space. In contrast, REMO22 employs joint probability distribution function methods.

The phases obtained from any MR procedure may not be of sufficient quality to confirm with certainty that the target structure has been solved. Therefore, it is common practice to refine the MR phases and use them as a starting point for AMB programs. To ensure a thorough and automated phasing process without any unanswered questions, we have developed three pipelines, REMO22 + SYNERGY + CAB, PHASER + SYNERGY + CAB and MOLREP + SYNERGY + CAB, which automate the MR process, phase refinement, and model building. This allows for a more efficient and complete phasing process, providing a more accurate view of the effectiveness of Molecular Replacement techniques for solving macromolecular crystal structures. Furthermore, we will also investigate the role of SYNERGY and CAB in the pipelines to determine whether they contribute significantly to the success of the phasing process or are simply trivial tools for refining phases and building models.

## 2. Results

In Section 2.1, we present the experimental results that demonstrate the effectiveness of REMO22 in solving MR problems. To assess its performance, we compared its results with those obtained by MOLREP and PHASER on the same set of test structures. In Section 2.2, we discuss the role of SYNERGY and CAB in the REMO22 + SYNERGY + CAB pipeline and compare its effectiveness with that of other pipelines.

### 2.1. About REMO22

A total of 157 macromolecular structures were used as test cases, comprising 101 proteins and 56 nucleic acids. The PDB codes of the test cases are listed in Table 1, and they are divided into five subsets: PH, PD, PG, DNA and RNA. The first three subsets contain proteins, while the last two contain nucleic acids. Further details can be found in the Section 4.

To keep Table 1 concise, we have not listed the molecular models used in the MR step. However, it is important to note that we used the models that were adopted for the original crystal structure solution, whenever possible. This was done to address the same problems that were encountered during the original solution process. It is possible that better models may be available today, which could make the solution easier. Unfortunately, for 37 of the 46 structures in the PG subset, we were unable to use this approach since they were solved using SAD-MAD techniques. Instead, we utilized search models obtained by Bond [60] by aligning the target and homologue sequence and by using the sequence alignment to trim and mutate the homologous chain with CHAINSAW [61]. To assist interested readers with their own reviews, all of the models used in this study are included in the Appendix A.

The extensive set of test structures listed in Table 1 was used to evaluate the effectiveness of REMO22 in default conditions for both proteins and nucleic acids across a wide range of scenarios. However, this evaluation cannot be considered complete without a comparison to the most popular MR programs available. Indeed, REMO22 should not be considered effective if it succeeds only in cases where another popular MR program succeeds and fails in cases where the same program fails. Therefore, we applied two of the most widely used and effective MR programs, PHASER and MOLREP, to the test structures listed in Table 1. We then compared the results obtained from these programs with those obtained by REMO22. In all our tests, we ensured that the same prior information was provided to all three programs, including measured reflections, space group, unit cell parameters, and MR models for the rotation and translation steps.

Given our focus on automating crystal structure solutions via MR techniques, we chose to apply each program using the default conditions, as suggested by the manuals. We recognize, however, that default procedures may not always be the optimal way to apply the software. Supplementary directives can alter default approaches and potentially increase the chances of a successful crystal structure solution. Despite this, default approaches are widely used by users and are typically the first choice. As the MOLREP default mode includes 20 restrained cycles of REFMAC to reduce the average phase error at the end of the MR step, we decided to add 20 REFMAC cycles to the PHASER automated MR mode. It is important to note that REFMAC cycles are already part of the REMO22 algorithms (see Section 4). Specifically, the PHASER run consisted of seven distinct steps: anisotropy correction, model generation, rotation function, translation function, packing function, rigid-body refinement and restrained REFMAC cycles. For MOLREP, we used its automatic mode, which represents an optimal balance between reducing CPU time and maintaining effectiveness. In this mode, anisotropy correction is not performed (like REMO22), but a packing function is included.

To evaluate the quality of the phases provided by REMO22, PHASER and MOLREP at the end of the MR procedure, some initial remarks are necessary.

Firstly, the procedure for locating copies of the model in the target asu is specific to each program. This includes the estimated number of model copies to locate, the rotation and translation search algorithms, and the FOMs used to rank the solutions. Secondly, a program may choose to simplify the MR techniques to save CPU time, while another program may invest in CPU time-consuming algorithms to improve the quality of the MR models. If two programs simultaneously fail or succeed, the program that saves CPU time is usually the preferred choice. However, if the program that requires more CPU time can solve more MR problems than the faster program, its procedure may be preferred. Therefore, any comparison between programs should consider the computer resources required by each algorithm.

The first figure of merit to evaluate the quality of the MR models, in the absence of any prior information on the target structure, is the final crystallographic residual R*,* which represents the accuracy of the model and influences the user’s trust in it. However, different programs define different resolution limits and subsets of phased reflections, so a fair comparison of R values can only be made when calculated over all observed reflections. Therefore, R values are calculated for each structure using the available MR models. To save time, we do not provide individual R values for each structure in this report, but they can be found in Appendix A section. It should be noted, however, that PHASER stops prematurely in five cases (2htt, 4gsg, 5i4s, 5lj4, 3fs0), when attempting to estimate the number of chains in the target asymmetric unit and produces an error message about the mismatch between composition and unit cell volume. Although user intervention can solve the problem, we treat these cases as failures of the automatic PHASER procedure for statistical purposes. For these cases, we assume an average phase error of 90°, and set the R value to 0.59, the expected R value for acentric random structures.

Table 2 presents a statistical analysis of the R values obtained by REMO22, PHASER, and MOLREP, based on the criteria described above. The final average R values for REMO22, PHASER and MOLREP, denoted as <R_R_>, <R_P_> and <R_M_>, respectively, were calculated for each subset of test structures.

The overall <R> values for PHASER and MOLREP are close to each other, at 0.43 and 0.40, respectively. In contrast, the <R> values for REMO22 are significantly smaller for each subset of test structures, indicating higher quality MR models and greater user trust in the program. Specifically, the overall <R> value of 0.34 for REMO22 suggests higher model quality compared to PHASER and MOLREP.

Interesting details in Table 2 are the NR30_R_, NR30_P_ and NR30_M_ entries, which indicate the number of cases in which each program produced an R value smaller than 0.30. Such cases represent high-quality MR models that do not require further refinement before being submitted to an AMB program. REMO22 produced models meeting this criterion in 61 cases, while PHASER and MOLREP did so in 22 and 36 cases, respectively.

The high-quality phases produced by REMO22 can also be demonstrated by the average phase errors, <*|*Δϕ*|*>_R_, <|Δϕ|>_P_ and <|Δϕ|>_M_, which represent the average deviation of the calculated phases from the published phases at the end of the MR step, for REMO22, PHASER and MOLREP, respectively. As with the R values, each <|Δϕ|> value does not refer to the reflection subset actively used in the MR step, due to the different MR resolution limits employed by the three programs. Instead, it relates to all the measured reflections and can therefore be regarded as an absolute, meaningful a posteriori figure of merit. In Figure 1, we present <|Δϕ|>_R_, <|Δϕ|>_P_ and <|Δϕ|>_M_, structure by structure, for each subset of test cases (i.e., PH, PD, PG, DNA and RNA). The structures are arranged in ascending order of <|Δϕ|>_R_ to facilitate readability. For interested readers’ numerical reference, we report <|Δϕ|>_R_, <|Δϕ|>_P_ and <|Δϕ>_M_ for each structure in Appendix A.

Insight into the overall quality of the REMO22, PHASER and MOLREP phases can be gained by examining the global average phase error calculated over all 157 test structures. Table 3 shows that REMO22 has the lowest average phase error, with <|Δϕ|>_R_ = 45°, followed by MOLREP with <|Δϕ|>_M_ = 56° and PHASER with <|Δϕ|>_P_ = 58°. These values are in good correlation with the corresponding <R> values presented in Table 2.

An additional criterion that may help readers in interpreting the experimental results presented in Figure 1 and in our tables is the use of the following rules of thumb.

If <|Δϕ|> is greater than or equal to 70° at the end of the MR step, it is highly likely that the MR model is either misplaced or inaccurate. In such cases, subsequent model refinement is likely to be unsuccessful or result in incomplete or rough models.

On the other hand, if <|Δϕ|> is less than 70°, the corresponding model is probably suitable for refinement, and the final AMB programs have a high probability of generating satisfactory structural models.

The cut-with-ridge criteria mentioned above are not absolute, as the success of model refinement depends on various factors such as data quality (e.g., |F|/σ(|F|) statistics, observed data resolution, percentage of solvent, and the effectiveness of the program used for phase refinement). However, using these criteria simplifies the analysis. Table 2 displays the number of test structures for which REMO22, PHASER and MOLREP exhibit a <|Δϕ|> ≥ 70° (N70_R_, N70_P_ and N70_M_, respectively). It is noteworthy that N70_R_ is significantly smaller than N70_P_ and N70_M_ for each subset of the test structures (22 against 55 and 50, respectively). For proteins, MOLREP seems to be more effective than PHASER, while PHASER appears to be more effective than MOLREP for nucleic acids (18 cases with <|Δϕ|> ≥ 70° against 24). The correlation of <|Δϕ|>_P_ and <|Δϕ|>_M_ with the <R_P_> and <R_M_> values presented in Table 2 suggests that the overall qualities of the structural models provided by MOLREP and PHASER are quite similar. Furthermore, the REMO22 structural models are of superior quality compared to those provided by PHASER and MOLREP.

Let us review Table 2 on the structure subsets. The subset PD presents the greatest difficulties due to the small SI values. In this case, REMO22 appears to be more effective than PHASER and MOLREP in limiting the adverse effects of SI, with only 7 cases of <|Δϕ|> ≥ 70° compared to 12 cases for PHASER and MOLREP. The PH subset, on the other hand, is generally easy to solve for all programs. However, the MR techniques are less effective for the PG subset than for the PH subset. The difficulty in PG is not due to smaller SI values, but rather to the number of model copies that need to be accommodated in the target asu, which is equal to or greater than 2 for 55% of the PG structures. MOLREP is particularly challenged in nucleic acids, with N70_M_ corresponding to approximately 43% of the nucleic acid test structures.

It is important to note that the better performance of REMO22 compared to PHASER and MOLREP is primarily due to the implementation of new algorithms (see Section 4) that require larger computer resources. REMO22 is the most demanding program in terms of CPU time, with PHASER and REMO22 requiring approximately 3 min and 4 h, respectively, if the CPU time for MOLREP is set to 1 min. This significant difference in CPU time is due to our decision to include a significant part of the phase refinement process in REMO22, which helps to identify the correct MR solution and also save CPU time in subsequent steps of the crystal structure solution process.

### 2.2. About the SIR22 Pipeline

As the title and content of this paper suggest, we aimed to develop an automated pipeline for solving crystal structures of macromolecules through MR techniques. However, our analysis of the experimental results obtained using REMO22, PHASER and MOLREP cannot be considered conclusive as the MR models were not subjected to model refinement and AMB, two essential steps in the crystal structure solution process.

To address this, we decided to submit the phases and weights obtained by these programs to the same refinement and AMB procedure using SYNERGY and CAB, respectively. SYNERGY’s efficacy was demonstrated by Burla et al. [39], while the ability of CAB was verified in a Paper III by Cascarano & Giacovazzo [56]. We implemented the three pipelines, REMO22 + SYNERGY + CAB, PHASER + SYNERGY + CAB and MOLREP +SYNERGY + CAB, into SIR22, a modified version of SIR2014 [62], for checking the automatic crystal structure solution via different MR techniques. The question we sought to answer was whether the SYNERGY and CAB modules add value to the MR programs or if most of the work was already done at the MR step, making SYNERGY + CAB a trivial bimodule for ending the phasing process.

Let us start with SYNERGY refinement. To simplify the analysis of our experimental results, we need to establish some criteria given the large number of test cases. The first criterion is to compare the average phase error <|Δϕ|>_MR_, calculated over all the test structures at the end of the MR step with the corresponding <|Δϕ|>_REF_, calculated after the SYNERGY phase refinement (see Table 3). The second criterion focuses on the number of cases where SYNERGY improves the MR average phase error by at least 10° (N_P_10 for proteins and N_NA_10 for nucleic acids), or by at least 20° (N_P_20 for proteins and N_NA_20 for nucleic acids).

Table 3 summarizes the statistical results for the segments REMO22 + SYNERGY, PHASER + SYNERGY, and MOLREP + SYNERGY, based on various criteria. We observe:i<|Δϕ|>_REF_ is consistently smaller than <|Δϕ|>_MR_, irrespective of whether SYNERGY is applied to the REMO22, PHASER, or MOLREP phases.iiREMO22 + SYNERGY provides the phases with the smallest average error (41°), while PHASER + SYNERGY and MOLREP + SYNERGY have average errors of 53° and 46°, respectively.iiiThe effectiveness of SYNERGY varies depending on the MR program. When applied to PHASER phases, SYNERGY provides an average phase improvement of 5°, whereas for MOLREP phases, it provides an improvement of 10°. However, for REMO22 phases, the improvement is only 4°. This is not surprising, as REMO22 phases are already refined phases (with an average phase error of 45°, compared to 58° and 56° for PHASER and MOLREP, respectively), making further refinement more challenging.ivThe number of test structures with a phase error reduction of more than 10° (N_P_10, N_NA_10) or 20° (N_P_20, N_NA_20) is much higher for the PHASER and MOLREP phases when SYNERGY is applied. Specifically, a reduction of more than 10° is observed for 10% of the test structures for the PHASER phases and 28% of the test structures for the MOLREP phases.vWe note that the larger effectiveness of SYNERGY for MOLREP phases compared to PHASER phases is not completely understood at this point.

It is possible that other refinement programs could yield better results than SYNERGY in improving the MR phases. To further investigate this issue, we decided to apply RESOLVE [63,64] as an alternative refinement program. RESOLVE is a highly respected package based on maximum-likelihood approaches [65,66,67] that expresses the experimental phase and amplitude information for a given structure factor in terms of a log-likelihood function and calculates the log-likelihood of the resulting electron-density map. Unlike SYNERGY, which employs traditional EDM techniques, RESOLVE assigns more realistic weights to the phases, thereby enhancing their effectiveness. If RESOLVE proves to be more effective than SYNERGY in improving MR phases, it could replace SYNERGY in the SIR22 pipeline, resulting in obvious benefits for the subsequent AMB step.

The results of the combination of PHASER + RESOLVE and MOLREP + RESOLVE are presented in the last two rows of Table 3. The following observations can be made:iRESOLVE leads to a 2° improvement in the PHASER and MOLREP phases, as compared to the 5° and 10° improvement obtained by SYNERGY, respectively.iiThe values of N_P_10, N_P_20, N_NA_10, N_NA_20 corresponding to RESOLVE phases are almost always close to zero. This means that RESOLVE is not able to improve the average phase errors by at least 10°, regardless of whether the phases were originally obtained by MOLREP or PHASER.iiiThe phases obtained by PHASER + RESOLVE are similar to those obtained by MOLREP + RESOLVE, making them an almost equivalent starting point for the application of the AMB programs.

Based on these observations, SYNERGY seems to be a more promising alternative to RESOLVE. Its significant phase improvements can be even more appreciated if one considers that there are cases in which the tested MR programs are not able to correctly locate the model and there are other cases in which the MR phase errors are already quite small. In both the above cases, it is unrealistic to hope for an improvement in the phase refinement step.

Let us now consider the role of CAB. Its potential was previously discussed in Papers II and III, where it was compared to BUCCANEER, NAUTILUS, ARP/wARP and PHENIX.AUTOBUILD, all run in their default settings. The results showed that the cyclic approach of CAB significantly enhances the effectiveness of BUCCANEER and NAUTILUS, and it is highly competitive with ARP/wARP and PHENIX.AUTOBUILD. With the larger set of protein structures analyzed in this paper, we can perform more meaningful tests.

One algorithm included in the current version of CAB is worth mentioning. In Paper III, we expanded the NAUTILUS library by adding representative structures of the A-DNA, B-DNA, Z-DNA, and four-stranded DNA forms. We also included the MR model because it was selected from structures with the highest sequence identity to the target structure and, by its nature, it deserves to be part of the library. In this version of CAB, we also added the MR model to the BUCCANEER library.

Let us begin by examining the three pipelines: REMO22 + SYNERGY + CAB, PHASER + SYNERGY + CAB and MOLREP + SYNERGY + CAB, to determine their success rate. Appendix A quotes the MA values (MA represents the percentage of non-H atoms within 0.6 Å of the published coordinates) obtained at the end of each pipeline for each test structure. However, for the sake of brevity and clarity, the user may be more interested in a shorter and more comprehensible statistical summary of the results. To accomplish this task, we adopted the following three criteria:iIf 65% or more of non-H atoms are within 0.6 Å of the published coordinates at the end of the CAB procedure, then the automatic crystal structure solution is considered successful. While some readers may find this percentage too lenient, and others too strict, we believe it to be practical, since refinement and completion of the model structure may be easily performed once this percentage is exceeded.iiIf less than or equal to 40% of non-H atoms are within 0.6 Å of the published coordinates at the end of the CAB procedure, then the automatic crystal structure solution fails.iiiPartial success occurs when a percentage smaller than 65% and larger than 40% is obtained.

Table 4 reports the number of structures with MA values lying in each interval (INT_MA_) for each pipeline.

We found that:iThe number of test structures for which the automatic crystal structure solution procedure succeeds, as per the criteria specified earlier, are: 122 for REMO22 + SYNERGY + CAB (N_RSC_), 93 for PHASER + SYNERGY + CAB (N_PSC_) and 108 for MOLREP + SYNERGY + CAB (N_MSC_). The failure cases, as per the same criteria, are 23 for REMO22 + SYNERGY + CAB, 52 for PHASER + SYNERGY + CAB, and 35 for MOLREP + SYNERGY + CAB.iiMOLREP phases resulted in a smaller number of CAB failures and a larger number of successes compared to PHASER. It is important to note that part of this bias is due to five cases where PHASER stops prematurely while trying to estimate the number of chains in the target asu. User intervention can solve this problem, leading to a statistical improvement in the PHASER results.


The quality of the molecular models provided by PHASER + RESOLVE + CAB (N_PRC_) and MOLREP + RESOLVE + CAB (N_MRC_) pipelines was also analyzed. Using RESOLVE instead of SYNERGY as the phase refinement program implies that:-13 structures are no longer automatically solved with PHASER data, while the number of failures increased by 17 (compare the columns N_PSC_ and N_PRC_).-14 structures are no longer automatically solved with MOLREP data, while the number of failures increased by 17 (compare the columns N_MSC_ and N_MRC_).

The results obtained indicate that the use of SYNERGY in the pipeline REMO22 + SYNERGY + CAB is not only effective, but it may also be beneficial in other pipelines that rely on different MR programs. However, it is important to note that the benefits of SYNERGY come at the cost of increased computing resources required by its algorithms. 

Furthermore, the individual contribution of CAB to the success of the REMO22 + SYNERGY + CAB pipeline can be assessed by replacing CAB with BUCCANEER for proteins and NAUTILUS for nucleic acids. It is worth noting that CAB essentially uses the same algorithms as BUCCANEER or NAUTILUS, but in a cyclic manner. To obtain the BUCCANEER or NAUTILUS results, the REMO22 + SYNERGY + CAB pipeline was stopped at the first cycle of the CAB procedure (as shown in Table 4). The number of structures automatically solved by the REMO22 + SYNERGY + (BUCCANEER or NAUTILUS) pipeline is 98 (NRS_BN_ in Table 4), which is 25 less than the number solved by the REMO22 + SYNERGY + CAB pipeline. However, the number of failures increases from 22 to 46. This demonstrates the significant contribution of CAB to the success of the pipeline.

In conclusion, the pipeline REMO22 + SYNERGY + CAB appears to be the most promising option among the tested pipelines. However, it also requires significant computer resources. To enable users of the pipeline REMO22 + SYNERGY + CAB to visually inspect the final structural models, a graphical program (JAV [68]) can be launched by the user. We are planning to automate this step in an upcoming release of SIR22. Figure 2 and Figure 3 show the JAV images of two structures, 3zyt with MA = 0.81, SI = 0.22, and 2i3p with MA = 0.63, SI = 0.99. We superimposed the CAB chains (in red) onto the chains corresponding to the published structures (in blue).

The two examples presented in Figure 2 and Figure 3 demonstrate that the CAB model can produce good overlap of chains with the published structures in some cases, while in others there may be significant deviations, even with a high SI value (as in the case of 2i3p) when parts of the structure are missing. Additional tests not included in this report indicate that an MA value of 0.65 is a reasonable threshold for a successful automatic crystal structure solution.

## 3. Discussion

The REMO09 algorithms for the MR step underwent significant modifications, and new algorithms were designed to create REMO22. This program is particularly suitable for the automatic crystal structure solution of biomolecules using MR techniques for both proteins and nucleic acids. To test the usefulness of REMO22 for the crystallographic community, we selected various proteins and nucleic acids and compared REMO22 results with those obtained by using PHASER and MOLREP. We chose the automatic approach recommended by the corresponding manuals for all three programs. The comparison of experimental results clearly indicates that the larger investment in terms of computing resources required by REMO22 is justified by a higher success rate when automatic approaches are used. Therefore, REMO22 can be considered a valuable alternative to the most used MR programs.

REMO22 is the first step in the REMO22 + SYNERGY + CAB pipeline, designed for automatic phasing using MR techniques. To understand the role of SYNERGY, we submitted the MR phases obtained from PHASER and MOLREP to RESOLVE, a popular phase refinement program. The results show that SYNERGY plays a crucial role in the success of automatic phasing procedures. Additionally, we tested the effectiveness of CAB by comparing it with BUCCANEER and NAUTILUS, and found that CAB significantly contributes to the success of the pipeline. Our findings suggest that investing more computer resources into the automatic crystal structure solution using MR techniques has led to the development of the REMO22 + SYNERGY + CAB pipeline, which is a valuable alternative to existing pipelines for solving the phase problem using MR techniques.

The comparison between the pipeline REMO22 + SYNERGY + CAB and the procedures for small-medium size molecules is instructive and raises the question of whether our pipeline can be considered an automatic crystal structure solution procedure similar to those available for small-medium sized molecules. While the main limits for small-medium molecules are the number of non-H atoms per asu and data resolution (300 non-H atoms per asu at 1.1 Å resolution are a hard limit for success unless enough heavy atoms are present), in the pipeline REMO22 + SYNERGY + CAB these parameters are not critical. More critical parameters are the sequence identity between model and target, the number of model copies to accommodate into the target asu, the presence of non-crystallographic symmetry, and the unknown crystal-chemical nature of the target chains.

Let us briefly discuss some of the reasons for failure in the protein structure determination process:iThe SI = 0.3 threshold presents a significant challenge, as evidenced by the fact that 4 out of 10 attempts failed (3nng, 3npg, 3nr6, 3tx8);iiThe inadequacy of the model used for protein complexes containing hetero-oligomers can lead to failure. For instance, the 1lat structure comprises two polypeptide chains of 71 and 74 residues, respectively, as well as two identical nucleic acid chains, each with 19 nucleotides. However, the model only corresponds to the polypeptide chains of the 1glu structure. Similarly, in the case of the 2iff structure, which is a complex of a monoclonal antibody (two chains of 212 and 214 residues), and a lysozyme (one chain of 129 residues), the model only contains the lysozyme chain of the 1hem structure. Even if the models are correctly positioned, recovering the full structure for these cases is a challenge;iiiInaccurate or incomplete prior information on the crystal-chemical nature of the target can also contribute to failure. For instance, DNA molecules are flexible and can adopt various structures, including G-quadruplex structures formed by nucleic acids rich in guanine. These structures are helical in shape but may be challenging to locate if the model is not a four-stranded DNA structure. Examples of G-quadruplex structures include 1s45, 1s47, 4wo3, and 5ua3;ivDisorder can also pose a challenge in determining protein structures. For example, in the cases of 3tok and 4gsg, each chain exhibits two distinct configurations, with most of the phosphorus atoms being common to both configurations. The relatively small MA values (0.45 and 0.41, respectively) are calculated with respect to the total number of atoms in the asymmetric unit, including the disordered pairs.

All of the aforementioned reasons, combined with the inherent statistical limitations of MR FOMs, caused REMO22 to completely fail in 23 cases. Despite this, REMO22 + SYNERGY + CAB must still be considered as a reliable and effective automated pipeline for solving crystal structures using MR techniques, as evidenced by its high success rate (122 out of 157). However, one significant drawback of the pipeline is its high CPU time requirement, which can be attributed to our implementation of new algorithms and the insufficient attention paid to computing times when connecting the three segments of the pipeline. Nonetheless, we are actively working on ways to significantly reduce the CPU time requirement in the near future.

## 4. Material and Methods

Burla et al. [39] used 24 protein structures out of 157 test cases to evaluate the SYNERGY refinement process of the phases obtained by REMO09. To increase the size of the test sample, this set was expanded to 40 (SET PH). The SET PD comprises 10 of the 13 structures investigated by DiMaio et al. [69] (which have experimental data available), characterized by an SI value smaller than 0.30. These structures were originally solved by combining PHENIX with ROSETTA, a suite [70] that uses physically realistic all-atom potential functions for predicting protein structures based on their amino-acid sequence. One of these structures (4e2t) has an SI of 1 and was used by DiMaio et al. to verify the method. Four test structures from SET PH (1cgn, 1cgo, 1e8a, 2f8m), for which SI < 0.40, were moved to SET PD. Additionally, we included 5ww0 in SET PD, a structure that was originally solved by a working version of REMO22 and has an SI of 0.23.

The SET PG consists of the remaining 46 protein test structures, which were deposited in the PDB by the Joint Centre for Structural Genomics, Wilson Laboratory, Scripps Institute. These structures are commonly used as a test case for MR studies.

For the nucleic acid structures, we selected 56 structures deposited in the PDB database (solved using MR techniques), thereby having observed diffraction data, unit cell information, space group symmetry, published sequences, and MR models available. Among these, 46 were used by Cascarano & Giacovazzo [56] as test cases to assess the effectiveness of the CAB approach for nucleic acid structures. The first 31 structures are DNA (SET DNA), and the remaining 25 structures are RNA fragments (SET RNA).

REMO09 utilized the method of joint probability distribution functions, which was adapted to different types of prior information. The same approach is maintained in REMO22, but several new algorithms have been incorporated to enhance the program’s robustness.

### 4.1. Extension to Nucleic Acids

REMO09 was originally designed to work only with protein structures. REMO22 has been extended to work with both DNA and RNA structures.

### 4.2. Estimation of the Number of Chains Per Asu and of the Number of Model Copies for MR

The current technique for estimating the number of chains in the target asymmetric unit (asu) is based on biochemical analysis, which establishes the size and sequence of macromolecular chains present in the target crystal structure. However, the actual number of chains per target asu is unknown. The most popular technique for estimating this number is the Matthews method [71], which is occasionally supplemented by considerations by Kantardjieff & Rupp [72], who found a correlation between solvent content and diffraction limits.

The Matthews method assumes implicitly that the protein chains in the unit cell have the same size and that the density of the protein (δ_prot_) is usually around 1.35 g/cm^3^, which is independent of the protein’s nature and molecular weight [73]. However, these assumptions are not always valid in practice. Fischer et al. [74] conducted tests that suggest that δ_prot_ = 1.41 g/cm^3^ is a suitable estimate for proteins with high molecular weight (i.e., M > 30 kDa). However, the protein density increases with decreasing molecular weight and reaches its maximum value of δ_prot_ = 1.50 g/cm^3^ for the smallest proteins (i.e., M ≈ 7 kDa).

Matthews’ survey of 116 different proteins suggests that the protein typically occupies 57% of the crystal volume, with occupancy values ranging from 75% to 35%. While the Matthews method works well in many cases, it can lead to ambiguity, especially for higher assembly numbers. A popular criterion for estimating the number of chains per target asymmetric unit (NCHT) is to choose the value that makes the protein volume fraction (PROTFRAC) closest to 0.50, a value estimated heuristically based on a large number of observations. This criterion is commonly used in PHASER, among other software tools.

While the early estimation of the target composition is not crucial for the success of MR, a more accurate early estimate can be beneficial, particularly when using an automatic approach. In REMO22, an algorithm is used to estimate the number of chains per asu and the number of model copies to accommodate in the target asu. The algorithm involves the following steps:
iIn small molecule crystallography, the expected number of molecules per asu is based on the volume per non-H atom (VOLAT), which is usually assumed to be between 16 and 18 Å^3^. For macromolecules, the sizes and sequences of the molecular chains present in a target crystal are typically known beforehand. However, the volume of the surrounding solvent remains unknown, making it challenging to estimate the number of chains per target asu. We have modified this rule based on a survey of a wide range of proteins and DNA-RNA structures. For proteins, the expected number of chains per target asu (NCHT) is that for which VOLAT is closest to 38 Å^3^, and not smaller than 22 Å^3^. For DNA structures, NCHT is that for which VOLAT is closest to 34.5 Å^3^, and not smaller than 22 Å^3^. For RNA structures, NCHT is that for which VOLAT is closest to 44 Å^3^, and not smaller than 22 Å^3^. The numerical values were established empirically.iiThe second step of the algorithm is aimed at estimating the number of model copies to accommodate in the target asu (NMOD). This information is typically sought after by the MR user. While not critical for the success of the MR procedure, a good early estimate of NMOD can simplify the automatic approach. Furthermore, this step can correct any incorrect NCHT estimate made in the first step of the algorithm. In cases where the model includes n identical chains, the NCHT value needs to be searched among multiples of n. However, there are scenarios where the target composition is made up of NCHT copies of two different sequence chains (one large and one small), while the model comprises only a single large chain. In such cases, confirming the experimental NCHT value is clearly incorrect, while NCHT/2 is a more accurate choice. Our algorithm can identify and address such situations, especially when the size of the smaller chain is insignificant compared to the larger chain (for example, less than 50% of the long chain). In such cases, the smaller chains are disregarded. The algorithm is designed to be flexible and can be applied to situations where the model and/or target consist of copies of chains of varying sizes.To assess the effectiveness of the choices mentioned above, we compared the number of incorrect estimates using the PROTFRAC criterion (50% solvent) versus the VOLAT criterion. Out of a total of 157 test cases, we discovered that the PROTFRAC criterion led to 30 erroneous NCHT estimates, whereas the VOLAT criterion resulted in only 15 incorrect estimates. These findings provide a promising foundation for the complete automation of the MR procedure. In addition, the NMOD value can be rectified in the third step of the algorithm, as described in the main text.iiiDuring the third step, it is possible to correct the number of model chains to be placed in the target asu through post-estimation. Let us assume that the orientation and location of the nth model have already been determined by the MR procedure, and that the figure of merit (FOM_n_) has been calculated to assess the reliability of the model’s position and orientation. The FOM_n_ value is expected to increase with the accuracy of the model, which corresponds to the number of accurately located model copies. If FOM_n+1_ is found to be less than FOM_n_, then the (n + 1)th copy of the model is rejected, the MR procedure is stopped, and the phase refinement step is started.

To evaluate the arrangement of the located chains, including symmetry-related copies, a second figure of merit, CLASH_n_, is calculated. For proteins, CLASH_n_ estimates the fraction of Cα atoms that overlap (within 3.0 Å) once the nth model has been located. For nucleic acids, it estimates the overlapping fraction of the phosphate and C atoms in the ribose-phosphate backbone and the N atoms of the bases.

Suppose we are assessing whether the (n + 1)th model copy should be accepted after the rotation and translation step. In that case, R(n) represents the crystallographic R-factor corresponding to the n located and accepted model copies, while R(n + 1) corresponds to the value related to the (n + 1) located copies. If R(n) − R(n + 1) > 0.02, the clash FOM is not checked and the (n + 1)th model copy is accepted. If CLASH_n+1_ > 35% or
R(n + 1) − R(n) > 0.15(1)
then the (n + 1)th model copy is rejected.

The meaning of the above conditions is clear. However, we have a supplementary condition: if
[R(n) − R(n + 1)]/CLASH_n+1_ > 0.10(2)
the (n + 1)th model copy is accepted, otherwise, it is excluded.

Let us examine the purposes of Conditions (1) and (2). If the (n + 1)th model copy is incorrectly oriented and/or located, Equation (1) is expected to be satisfied, and the rejection of the (n + 1)th model copy is warranted. In cases where R(n + 1) − R(n) is positive but very small, and CLASH_n+1_ is sufficiently large, it may be risky to include the (n + 1)th model copy in the current model. Conversely, if CLASH_n+1_ is very small, and R(n + 1) − R(n) is also sufficiently small to meet Condition (2), accepting the (n + 1)th model copy appears to be a reasonable decision. To avoid numerical divergence in Equation (2), we consider a CLASH value below 0.10 to be insignificant. Therefore, if CLASH < 0.05, we set CLASH to 0.05 in Equation (2). This algorithm is applied identically to both proteins and nucleic acids.

### 4.3. Resolution Limits

The subsets of reflections used in the rotation and translation steps are chosen automatically. Reflections with a resolution of up to 7 Å are excluded from calculations, except in situations where SI is less than 0.5. The maximum accepted resolution for active reflections is 2.5 Å, and reflections with very high or very low normalized structure factor moduli are also disregarded. The SI value is not considered for nucleic acids, mainly because nucleic acid helices can assume comparable conformations, even when their sequences are substantially different.

### 4.4. Search Algorithm for the Rotation Step

The orientation space is based on the asymmetric region of the rotation group [75]. First, the atomic coordinates of the model are orthonormalized, and the maximum molecular dimension is calculated. Then, an orthogonal reciprocal lattice grid is generated, with the direct space dimensions chosen to be four times the maximum molecular dimension. The model is rotated by rotating the observed reciprocal lattice with respect to the model lattice, and the structure factors of the molecular model are calculated only once.

To rotate the model, an angular grid dθ is used, with n_1_dθ, n_2_dθ, n_3_dθ being the Euler angles corresponding to the cubic primitive lattice. There is at least one point in the unit cell of such a lattice that is approximately 0.87dθ from the lattice points (i.e., the center of the cubic cell). To reduce sampling errors, the angular grid can be lowered to dθ/2, but this results in eight times more lattice points. An alternative approach is to explore the orientation space using a body-centered lattice, which doubles the number of lattice points but ensures that no point in the body-centered cubic cell is farther than 0.56dθ from any lattice point. The body-centered cubic lattice is obtained by first exploring the orientation space using a primitive lattice and then exploring the same angular space using the same primitive cubic lattice, but starting from (dθ/2, dθ/2, dθ/2).

### 4.5. Anisotropy Correction

Anisotropy in diffraction data refers to the fact that diffraction intensities decrease at different rates in different directions of the reciprocal lattice. As a result, the FOM criteria used to select the correct solution in MR may fail. The reason for this is that the observed diffraction intensities are often anisotropic, while the calculated intensities, particularly in the early stages of the process, are usually assumed to be isotropic. To overcome this limitation, it is necessary to make the calculated and observed structure factors thermally homogeneous. This can be achieved by renormalizing the normalized structure factors according to their direction before calculating the FOMs. To estimate the degree of anisotropy, one can examine how the overall principal components of the anisotropic atomic displacement parameters vary in different directions of reciprocal space. In REMO22, a mathematical approach based on previous work on the preferred orientation of crystallites in a powder [76] is applied to account for anisotropy in the diffraction data.

Let us consider a scenario where the normalized structure factor moduli, |*E*|, have been calculated, and n reciprocal lattice points (with n being approximately 30) have been selected, which correspond to n directions [**h**] = [hkl]. If these points are chosen at very low resolution (e.g., [100], [010], [001], [110], [101], etc.), they will represent all directions in reciprocal space and will be referred to as polar directions. For each polar direction [**h**], the following steps are executed:(1)The reciprocal space is divided into cones, all with the same axis as the polar direction. The cones are arranged so that each one is fully contained in the next. The shells (i.e., the regions of reciprocal space between adjacent cones) have approximately equal volumes and therefore contain approximately the same number of lattice points. For each shell, α is the average angle (**k**, **h**), where **k** is the generic lattice point in the shell.(2)For each shell, <|*E***_k_**|^2^> is calculated, and the corresponding values are plotted against α.(3)The von Mises distribution
M = exp (G cos 2α)
is found, where G is the parameter best fitting the experimental <|*E***_k_**|^2^> distribution. If G is large and positive, then <|*E***_k_**|^2^> > 1 along the **h** direction, if G is large and negative then <|*E***_k_**|^2^> < 1 along the **h** direction.

Assuming that steps 1–3 have been applied to all n polar axes, if the values of all the G’s are close to zero, then the reciprocal space is nearly isotropic. However, if some G’s (either positive or negative) are significantly large, then the reciprocal space is mainly anisotropic.

To correct for anisotropy, one can easily represent the overall anisotropy of the reciprocal space with an ellipsoid. The geometrical shape of the ellipsoid depends on the crystal system being studied: it is spherical for the cubic system, a two-axis ellipsoid for the trigonal, hexagonal and tetragonal systems, and a three-axis ellipsoid for the orthorhombic, monoclinic and triclinic systems.

The orientation of an ellipsoid in reciprocal space is influenced by its underlying symmetry. In trigonal, hexagonal, and tetragonal systems, one of the two ellipsoid axes must align with **c***. In contrast, the orthorhombic system requires all three ellipsoid axes to be parallel to **a***, **b*** and **c***. In monoclinic systems, one of the three ellipsoid axes aligns with the unique two-fold axis **b***. In the absence of symmetry constraints, the ellipsoid orientation in triclinic systems is not predetermined.

To illustrate why these constraints exist, consider the orthorhombic system. The directions **a***, **b*** and **c*** are unrelated by symmetry elements, and the ellipsoid must have three axes to account for all possible anisotropy values against the crystal symmetry. To avoid discrepancies, the three ellipsoid axes necessarily align with **a***, **b*** and **c*** because otherwise the [hkl], [-h-kl], [-hk-l], [h-k-l] directions should have different anisotropy values with respect to the crystal symmetry.

Let us examine how to correct the anisotropy of the reciprocal space. If the G value is sufficiently large for certain polar directions, a correction parameter *O* can be calculated for each reflection, considering the crystal symmetry.

For hexagonal-trigonal systems:*O*(hkl) = <*E*^2^>_[100]_ (cos^2^ ϑ_1_ + cos^2^ ϑ_2_ ) + <*E*^2^>_[001]_ cos^2^ ϑ_3_
where ϑ_1_ is the angle between the direction [hkl] and the direction [100], ϑ2 is the angle between [hkl] and [1−20], ϑ3 is the angle between [hkl] and [001]. E denotes the normalized structure factor corresponding to *F*.

For the tetragonal system:*O*(hkl) = <*E*^2^>_[100]_ (cos^2^ ϑ_1_ + cos^2^ ϑ_2_) + <*E*^2^>_[001]_ cos^2^ ϑ_3_
where ϑ_1_, ϑ_2_, ϑ_3_ are the angles between [hkl] and [100], [010], [001] respectively.

For the orthorhombic system:*O*(hkl) = <*E*^2^>_[100]_ cos^2^ ϑ_1_ + <*E*^2^>_[010]_ cos^2^ ϑ_2_ + <*E*^2^>_[001]_ cos^2^ ϑ_3_
where ϑ_1_, ϑ_2_, ϑ_3_ are the angles between [hkl] and [100], [010] and [001] respectively.

Two or three measurements (depending on the system) are enough to define the ellipsoid.

The monoclinic system requires additional calculations due to its unique symmetry. One of the three ellipsoid axes aligns with the direction [010], while the other two must be selected in the plane defined by **a*** and **c***. As a result, these axes coincide with the directions [h0l]. To correct for anisotropy in the monoclinic system, the polar direction with the largest G value, denoted as [h_1_0l_1_], is identified. Next, a direction [h_2_0l_2_] that is approximately or exactly perpendicular to [h_1_0l_1_] is sought. This direction will be used to correct the anisotropy of the reciprocal space:$$O\left(\mathrm{hkl}\right)=\langle E^{2}\rangle {}_{\left[h_{1}0{\;l}_{1}\right]}{\;\cos}^{2}\;\vartheta_{1}+\langle E^{2}\rangle {}_{\left[010\;\right]}\cos^{2}\;\vartheta_{2}+\langle E^{2}\rangle {}_{\left[h_{2}0{\;l}_{2}\right]}{\;\cos}^{2}\;\vartheta_{3}$$
where ϑ_1_, ϑ_2_, ϑ_3_ are the angles between [hkl] and [h_1_0l_1_], [010] and [h_2_0l_2_], respectively.

To correct for anisotropy in the triclinic system, the following procedure is applied. First, the polar direction with the largest G value, denoted as [h_1_k_1_l_1_], is identified. Next, a direction [h_2_k_2_l_2_] with the largest G value is found in the plane that is approximately or exactly perpendicular to [h_1_k_1_l_1_]. Finally, a direction [h_3_k_3_l_3_] is identified that is perpendicular to both [h_1_k_1_l_1_] and [h_2_k_2_l_2_], either exactly or approximately. These directions will be used to correct the anisotropy of the reciprocal space in the triclinic system:$$O\left(\mathrm{hkl}\right)=\langle E^{2}\rangle {}_{\left[h_{1}k_{1}l_{1}\right]}{\;\cos}^{2}\;\vartheta_{1}+\langle E^{2}\rangle {}_{\left[h_{2}k_{2}l_{2}\right]}\cos^{2}\;\vartheta_{2}+\langle E^{2}\rangle {}_{\left[h_{3}k_{3}l_{3}\right]}{\;\cos}^{2}\;\vartheta_{3}$$
where ϑ_1_, ϑ_2_, ϑ_3_ are the angles between [hkl] and [h_1_k_1_l_1_], [h_2_k_2_l_2_] and [h_3_k_3_l_3_] respectively.

The anisotropy is then corrected by calculating the renormalized structure factors according to
|*E*′|^2^_obs_ = |*E*|^2^_obs_/*O*
which replace the |*E*|^2^_obs_ in the RFOM calculations.

### 4.6. Figures of Merit for the Rotation Step

Giacovazzo [77] developed a method to directly derive the conditional probability distribution of a structure factor based on different types of prior information without calculating the joint probability distribution functions.

Once n model copies have been oriented and placed, the orientation of the (n + 1)th model copy in the target asu can be determined using the *RFOM* figure of merit, where
(3)$$RFOM=CORR\left({\left|F\right|}^{2},\;\langle {\left|F\right|}^{2}\rangle \right)\;$$

*RFOM* is the correlation between $|F{|}^{2}$ and the expected value
(4)$$\langle F^{2}\rangle =|F_{p1}+F_{p2}+\ldots F_{pn}{|}^{2}+\overset{m}{\underset{s=1}{\sum }}|F_{ps}{|}^{2}\;$$
where *F_p_*_1_, *F_p_*_2_, …, *F_pn_* are the structure factors corresponding to the first, second,…, *n*th located model copy, *m* is the number of symmetry operators for the given space group and $\sum_{s=1}^{m}|F_{ps}{|}^{2}$ refers to the (*n* + 1)th model copy, for which we are searching the correct orientation. When *n* = 0, meaning that the first model copy is being rotated, Equation (4) simplifies to:(5)$$\langle |F{|}^{2}\rangle =\overset{m}{\underset{s=1}{\sum }}|F_{ps}{|}^{2}$$

The *RFOM* figure is designed to identify the orientation of the (n + 1)th model copy that maximizes the *RFOM* value, which is expected to correspond to the correct solution. When searching for the orientation of the first model copy, the 200 orientations that correspond to the highest *RFOM* values are selected and passed to the translation step.

### 4.7. Figures of Merit for the Translation Step

A preliminary selection of the most promising translation vectors is made by using the criterion
(6)$$\sum_{h}\left|F_{h}{|}^{2}\right|F_{ph}{|}^{2}=max$$

The left-hand side of Equation (6) is calculated via Fast Fourier Transform techniques according to Vagin & Teplyakov [78]. For each selected rotation, up to two translation vectors are accepted. However, the final ranking of the translation vectors is not determined immediately, as Criterion (6) may fail due to the imperfect orientation of the molecule, the presence of intermolecular vectors mixed with intramolecular ones, and the small sequence identity between the model and target. As a result, the determination of the final ranking of the translation vectors is postponed until these issues can be addressed.

Supplementary steps are taken to improve the ranking before making the final selection. First, the selected translations are scored [77] based on the following criterion
(7)$$TFOM=CORR\left(\left|F\right|,\left|F_{p}\right|\right)\;$$
where
(8)$$\left|F_{p}\right|=\left|F_{p1}+F_{p2}+\ldots F_{pn}\right|$$

*F_pi_* represents the structure factor of the model, which is calculated based on the position of the i-th previously located copy of the model. When *n* = 0, meaning that the first model copy is being rotated, Equation (8) simplifies to:(9)$$\left|F_{p}\right|=\left|F_{p1}\right|$$

Criterion (9) benefits from a typical statistical behavior: when one or more model copies have already been placed, the variance representing the uncertainty in locating the next components is reduced. This results in an increase in the ratio of signal-to-noise for the next components.

However, Equation (7) may not work effectively when high-resolution data have been measured. In this case, the molecular model, which is defined by rotation and translation parameters based on low-resolution reflections (usually between 3 and 4 Å), may not be of sufficient quality for the high-resolution reflections. Small errors in these parameters may lead to large errors in the calculated amplitudes and phases of the high-resolution reflections. In this case, we still rely on Equation (7), but TFOM is calculated only on the reflections which are actively used in the MR step. 

A situation where Equation (7) may not work effectively is when there is a pseudo-translational symmetry present. This type of symmetry generates a group of reflections with high intensities and another group with low intensities. To address this issue, Equation (10) is employed, where
(10)$$TFOM=1-<{\left|E_{p}\right|}^{2}>=max$$
$\langle |E_{p}{|}^{2}\rangle$ is determined by computing it for reflections where the normalized observed structure factor |*E*| is less than 0.3. Here, *E_p_* is the normalized structure factor of *F_p_*.

In our experience, both Criteria (7) and (10) are effective scoring functions. However, in some cases, the model may not be accurate enough, or the data may have limitations, making it difficult to identify good solutions based solely on the score values. To overcome such challenges, we select a variable number of the most promising solutions, selected by using either Criterion (7) or (10): they are further refined by using a rigid body refinement technique called SIMPLEX (see Section 4.8).

It is worth noting that RFOM and TFOM are unweighted FOMs, and we have not found any meaningful weights that can make them more effective.

### 4.8. Rigid Body Refinement by SIMPLEX

The solutions identified by the FOMs described in Section 4.7 are refined using the SIMPLEX method [79], which is an unconstrained optimization technique related to the downhill method. Here, the SIMPLEX method is applied to a six-dimensional parameter space, with three dimensions for rotation and three for translation. The refinement process typically results in a smaller average phase error, and it also facilitates the clustering of closely related solutions.

### 4.9. Selection of the Correct Solutions

The solutions refined using the SIMPLEX method undergo a cyclic procedure that combines applications of EDM and REFMAC [80]. This procedure is primarily focused on phase extension and refinement, and is crucial for the success of the crystal structure determination process. During this step (referred to as PRESYN to indicate that it precedes the SYNERGY step), the rigid body model obtained from the MR step is transformed into a model where individual atoms can shift to new positions under the control of REFMAC restraints. This cyclic procedure typically lowers the average phase errors of correct solutions while leaving the errors of false solutions unchanged. This makes it easier to distinguish correct solutions from false ones. The best solution is then identified based on the minimum REFMAC R value.

If only one copy of the model needs to be placed, the best solution is passed to the SYNERGY step for final phase refinement, and then to CAB. If multiple copies of the model need to be located, the top five solutions are selected, and, one at a time, each is used as prior information to locate the second copy. The same practice is used to locate additional copies of the model (see Section 4.10).

### 4.10. About the Location of the Second and Further Model Copies

Let us consider a scenario where the first copy of the model has been oriented and located, and the search for the second model copy’s rotation has started. REMO09 recovers the three Euler angles and corresponding three shift vectors that define the orientation and translation of the first model copy, and the *Fp*_1_ values are calculated to start the search for the second model copy’s roto-translation. However, this approach has a potential pitfall because *Fp*_1_ arises from a rigid body model, and inaccuracies in the model orientation and location, as well as structural differences between the model and target, may create a systematic bias that can affect the FOMs effectiveness. As a result, it can be challenging to recognize the correct roto-translation parameters for the second model copy.

To overcome this issue, REMO22 refines for the first model copy using REFMAC, causing it to lose its original rigidity. Accordingly, structure factors corresponding to the first model copy are calculated from appropriate coordinates and used as prior information to locate the second model copy. The resulting phase improvement makes FOMs more effective at identifying the correct orientation and position of the second model copy. The same approach is used to locate additional copies. When all model copies are located, only the best solution is submitted to SYNERGY.

### 4.11. Automatic Restart

The success rate of MR may decrease when it is applied to models with lower scattering power compared to the target asu or when the root mean square deviation between the model and target structures is large. Let us assume that the final R value at the end of CAB is too high for proteins with SI < 0.4. In such cases, REMO22 is automatically restarted using a different strategy. According to Chothia & Lesk [81], when SI = 0.4, the root mean square deviation from the correct positions is 1.22 Å, which is likely an underestimate. This value makes it challenging to identify the correct rotation and translation. In these circumstances, it is expected that a high value of the crystallographic residual R will be observed between the calculated and observed structure factors, even when the model is correctly located. In REMO22, as is already the case in REMO09, when the SI < 0.4, up to 80% of the residues with the largest isotropic temperature factor are routinely treated as alanine during the SYNERGY step. This is done in the hope of removing atoms that are too far from their correct positions in the model. If the final R value at the end of CAB is greater than 0.50 and the SI < 0.4, a fully “alaninized” model is resubmitted to the REMO22 procedure.

### 4.12. Essential Directives

The full REMO22 + SYNERGY + CAB pipeline can be run automatically with very few directives. As an example, we will use 1aki structure:


*%cab buccaneer*



*%structure 1aki*



*%job ORTHORHOMBIC FORM OF HEN EGG-WHITE LYSOZYME AT 1.5 Å RESOLUTION*



*%data*



*mtz 1aki.mtz*



*label H K L F SIGF*



*sequence 1aki.seq*



*%remo*



*fragment 2ihl.pdb*



*%end*


If the users prefer to use PHASER or MOLREP as an MR program, they will need to provide a few additional directives to process their data through the segments SYNERGY + CAB (see the Appendix A section).

## Figures and Tables

**Figure 1 ijms-24-06070-f001:**
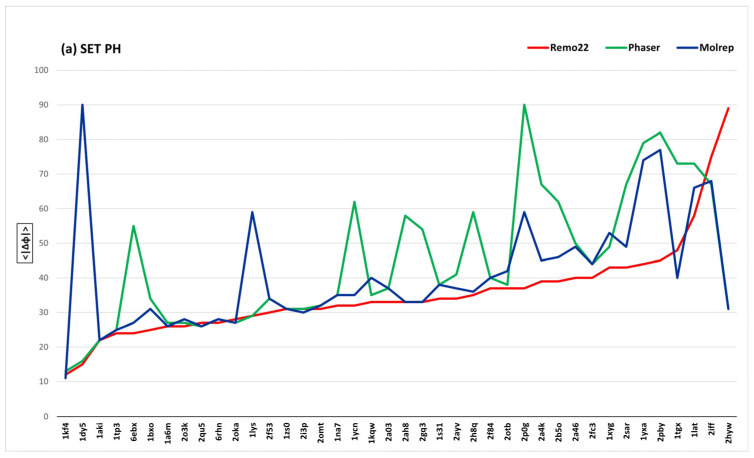
The average phase errors <|Δϕ|> (in degrees) obtained by REMO22 (<|Δϕ|>_R_; red line), PHASER (<|Δϕ|>_P_; green line) and MOLREP (<|Δϕ|>_M_; blue line) at the end of the MR steps ((**a**) SET PH; (**b**) SET PD; (**c**) SET PG; (**d**) SET DNA; (**e**) SET RNA). In cases where an MR program declares a failure before the standard ending, we assume <|Δϕ|> = 90° (in five DNA-RNA cases for PHASER). The structures are ordered in increasing values of (<|Δϕ|>_R_ ) for clarity.

**Figure 2 ijms-24-06070-f002:**
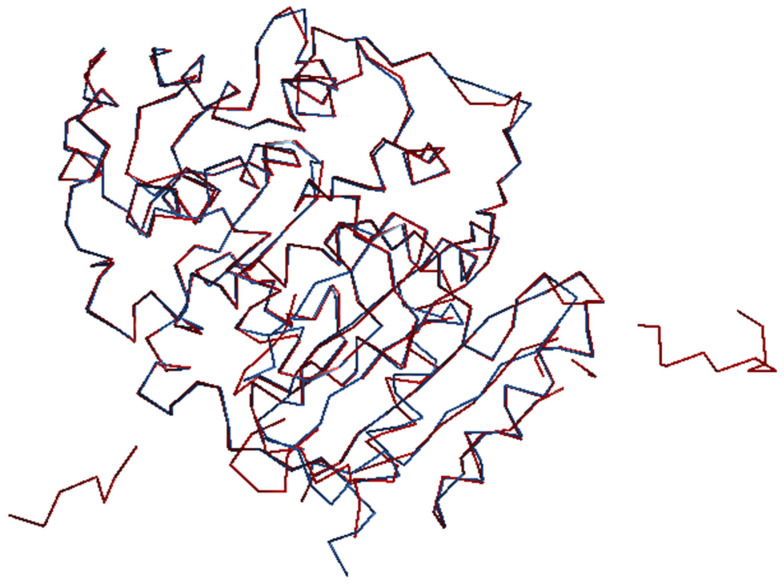
3zyt*,* MA = 0.81, SI = 0.22. CAB chain-trace in red, published chain-trace in blue. CAB and the published backbones coincide in most of the target asu.

**Figure 3 ijms-24-06070-f003:**
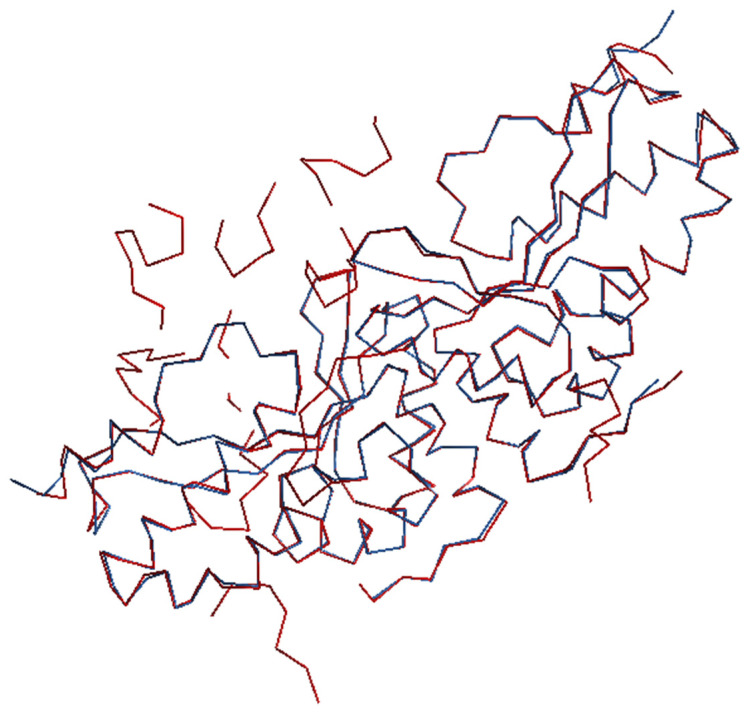
2i3p, MA = 0.63. CAB chain-trace in red, published chain-trace in blue of the protein component. There are regions of the target asu in which CAB and the published backbones do not coincide.

**Table 1 ijms-24-06070-t001:** PDB codes of test structures for MR applications, organized by set: PH, PD, PG, DNA and RNA.

SET	PDB	PDB	PDB	PDB	PDB	PDB	PDB	PDB	PDB	PDB
**PH**	1a6m	1aki	1bxo	1dy5	1kf4	1kqw	1lat	1lys	1na7	1s31
	1tgx	1tp3	1xyg	1ycn	1yxa	1zs0	2a03	2a46	2a4k	2ah8
	2ayv	2b5o	2f53	2f84	2fc3	2gq3	2h8q	2hyw	2i3p	2iff
	2o3k	2oka	2omt	2otb	2p0g	2pby	2qu5	2sar	6ebx	6rhn
**PD**	3nng	3npg	3nr6	3o8s	3on5	3q6o	3tx8	3zyt	4e2t	4fqd
	1cgn	1cgo	1e8a	2f8m	5ww0					
**PG**	1vkf	1vki	1vl2	1vl7	1vlc	2wu6	2x7h	3e49	3gp0	3h9e
	3h9r	3khu	3l23	3llx	3m7a	3mbj	3mcq	3mdo	3mz2	3nyy
	3obi	3oz2	3p94	3ufi	3us5	4e2e	4ef2	4ezg	4fvs	4gbs
	4gcm	4ler	4mru	4ogz	4ouq	4q1v	4q34	4q53	4q6k	4q9a
	4qjr	4qni	4r0k	4rvo	4rwv	4yod				
**DNA**	1s45	1s47	2b1d	2htt	3ce5	3eil	3gom	3goo	3n4o	3tok
	4gsg	4l24	4ltl	4ms5	4wo3	4xqz	4zym	5cv2	5i4s	5ihd
	5j0e	5ju4	5lj4	5mvt	5nt5	5t4w	5tgp	5ua3	6f3c	6h5r
	6tzq									
**RNA**	1iha	1lc4	1mwl	1q96	1z7f	2a0p	2fd0	2pn4	3d2v	3fs0
	3owi	3oxm	3s49	3td1	4enc	4jab	5fj0	5kvj	5l4o	5nz6
	5ux3	5uz6	5zeg	6az4	6cab					

**Table 2 ijms-24-06070-t002:** Performance comparison of REMO22, PHASER and MOLREP (the subscripts R, P and M represent the three MR programs) The final average R values (in %), denoted by <R_R_>, <R_P_> and <R_M_> respectively, were calculated for each subset of test structures. NR30_R_, NR30_P_ and NR30_M_ are the number of test structures for which the final R value was ≤0.30. N70_R_, N70_P_, and N70_M_ are the number of test structures for which the final <|Δϕ|> value of ≥70°.

SUBSET	<R_R_>	<R_P_>	<R_M_>	NR30_R_	NR30_P_	NR30_M_	N70_R_	N70_P_	N70_M_
PH	30	36	31	24	16	20	2	5	3
PD	42	50	50	1	0	0	7	12	12
PG	35	43	38	14	3	7	6	20	11
DNA	34	45	46	11	3	2	3	10	16
RNA	34	46	42	11	2	7	4	8	8
OVERALL	34	43	40	61	22	36	22	55	50

**Table 3 ijms-24-06070-t003:** The results for each pipeline segment are quoted: (i) the global average phase error ${\left\langle |\Delta \phi |\right\rangle }_{\mathrm{MR}}$ calculated over all the test structures at the end of the MR step via REMO22, PHASER and MOLREP, and the corresponding ${\left\langle |\Delta \phi |\right\rangle }_{\mathrm{REF}}$ calculated after the phase refinement step using either SYNERGY or RESOLVE. All phase errors are in degrees; (ii) the number of proteins for which SYNERGY or RESOLVE improves the MR average phase error by at least 10° (N_P_10) and by at least 20° (N_P_20); (iii) the number of nucleic acids for which SYNERGY or RESOLVE improves the MR average phase error by at least 10° (N_NA_10) and by at least 20° (N_NA_20).

Pipeline Segment	${\left\langle |\Delta \phi |\right\rangle }_{MR}$	${\left\langle |\Delta \phi |\right\rangle }_{REF}$	N_P_10	N_P_20	N_NA_10	N_NA_20
REMO22 + SYNERGY	45	41	12	6	0	0
PHASER + SYNERGY	58	53	16	6	17	4
MOLREP + SYNERGY	56	46	44	23	16	10
PHASER + RESOLVE	58	56	1	0	0	0
MOLREP + RESOLVE	56	54	1	0	1	0

**Table 4 ijms-24-06070-t004:** MA denotes the percentage of non-hydrogen atoms within 0.6 Å of the published atomic coordinates, represented by the metric MA. The number of structures (NRSC, NPSC, NMSC, NPRC, NMRC, NRSBN) with MA belonging to each MA interval (INTMA) are shown *.

INT_MA_	N_RSC_	N_PSC_	N_MSC_	N_PRC_	N_MRC_	N_RSBN_
MA > 65	122	93	108	80	94	98
40 < MA ≤ 65	12	12	14	8	11	13
MA ≤ 40	23	52	35	69	52	46

* The entries in the table are generated by six pipelines, namely REMO22 + SYNERGY + CAB (N_RSC_), PHASER + SYNERGY + CAB (N_PSC_), MOLREP + SYNERGY + CAB (N_MSC_), PHASER + RESOLVE + CAB (N_PRC_), MOLREP + RESOLVE + CAB (N_MRC_) and REMO22 + SYNERGY + (BUCCANEER or NAUTILUS) (N_RSBN_).

## Data Availability

The data are present within the article.

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
