# Peer review of "The Automatic Solution of Macromolecular Crystal Structures via Molecular Replacement Techniques: REMO22 and Its Pipeline"

_ijms, 2023, doi:10.3390/ijms24076070_

Round 1

Reviewer 1 Report

The article by Carrozzini et al. describes the updates implemented into REMO09, now REMO22, a pipeline for phase retrieval using molecular replacement (MR). The pipeline is then applied to a large selection of protein and nucleic acid structures deposited in the PDB and compared to the incumbent MR programs MOLREP and Phaser.

In a next step, the REMO22 pipeline is expanded into a further pipeline that “includes” phase refinement and automatic model building (SIR22). MOLREP and PHASER are again substituted in for REMO22 and outcomes compared.

The materials and methods section is thorough, and conclusive and provides a detailed description of the upgrades implemented to upgrade from REMO09 to REMO22.

The first part of the results on the other hand (the comparison with MOLREP and PHASER) is clumsy at best. When REMO22 by itself is compared to MOLREP and PHASER (run using “default” settings) the phase error for the former is so much better than for the other 2 programs, that the only logical assumption is that some phase refinement must already be implemented in REMO22, and that the comparison being made is between apples and oranges. Surprisingly (or unsurprisingly?) this is the conclusion made by the authors at the end of the comparison, so this reviewer is baffled by why 7 pages of results are dedicated to a comparison that makes no sense to begin with. If it solely for making a case that REMO22 outperforms MOLREP and PHASER, this is misleading since it is not a just comparison in the way it is presented. Most interestingly, REMO22 uses REFMAC for phase refinement. Noting that the “default” setting when running recent versions of MOLREP through the ccp4 suite is to finish off with shift field refinement followed by 20 cycles of  restrained refinement (with REFMAC!), and that this would make for a worthy comparison to REMO22, the authors choose to deviate from this particular “default” setting (maybe using an old version of MOLREP?). The option of running REFMAC on top of the output solution in basic PHASER is also an option implemented directly into PHASER, albeit indeed not a “default” setting in this case. It is therefore recommended that either the refined phases output by REMO22 be compared with refined phases output by MOLREP and PHASER, or this comparison is moot.

The final part of the paper integrates the REMO22 pipeline into the SIR22 pipeline which includes (additional) phase refinement, model refinement and model building. Surely the authors are aware that the naming convention of an (as it is portrayed in the current manuscript) MR-based pipeline as “SIR” is very odd and confusing to the general reader, considering the abbreviation SIR is commonly used for de novo “single isomorphous replacement” protein phase retrieval. I believe SIR2014 does also include true SIR phasing techniques as well as MR. A dedicated pipeline for MR (spliced from SIR2014 with a more suitable name) that includes only the MR methods may be more palatable to a reader who has just spent a manuscript reading only about MR. Since the pipeline is REMO22/MOLREP/PHASER + SYNERGY + CAB, is the introduction of a new SIR22 version necessary for this particular manuscript? Alternatively, moving the integration of REMO22 into SIR22 to a 2nd separate manuscript focusing on that particular pipeline’s capabilities and upgrades is suggested, as it’s less confusing to the reader.

Minor comments:

The detail to which Alphafold’s structure predictions is discussed in the introduction are reminiscent of a review article and not a methods paper on MR (that does not make direct use of Alphafold predictions per se). That structure predictions will always need experimental data to be corroborated is not under dispute, but the last paragraphs of the introduction reads as though it were and puts the importance of developing new MR programs into question. This is contrary to what is the case and what I believe the authors are trying to relay; i.e. that Alphafold benefits MR phasing (not necessarily needing to resort to more difficult and tedious experimental phasing techniques for unknown structures) but will never replace it. This reviewer suggests removing the (extensive) list of Alphafold limitations from the introduction, and instead adding a paragraph about how MR phasing can/will benefit from the largely accurate Alphafold predictions to the conclusion/outlook section of the manuscript.

Table 4 promises the reader results from 7 pipelines. But only results from 6 are presented.

Software versions of all named packages would be appreciated.

Abbreviations also need to be introduced in the text their respective first occurrence, and not just listed at the end of the methods.

Language can be improved: Sentence structure is a bit out of order throughout, and sometimes a bit more characteristic of a novel. E.g. Page 6 l228 : “Let us now discuss [..]”

When listing things, should be “a, b, c AND d.” not “a, b, c, d.” e.g. page 1 l30; page 2 l67; page 2 l71; page 3 l121; page 4 l193;  

It took a second to understand that PH, PD and PG are not some undefined abbreviations, but the names given to the protein dataset groups. This should be clarified.

Page 6 &7 “R’s” should be R-values. There’s no possessive.   

Figure 1: Legend missing and caption is above image instead of below.

Author Response

We thank the reviewer for the critical reading of the text.

Reviewer 2 Report

Comment

Date: 30-01-2023

Manuscript ID: IJMS-2203648

Carrozzini et al. addressed various findings in their research article entitle as “The automatic solution of macromolecular crystal structures via molecular replacement techniques: REMO22 and its pipeline”. The work is interesting and informative to reader working in the domains. However, I recommend major suggestions before publication.

Comment 1: In introduction, the sentence “Success requires that at least one of the following two serious conditions be met: atomic or quasi atomic data resolution, or sufficiently heavy atoms present in the unit cell.” is confusing. Please rewrite to make it clear and concise.  Paragraph first and second must be combined. What do you mean by “the last authors”?.  There is no reference cited for the sentence “The last authors were able to solve at non-atomic resolution large size protein structures (for example, 1e3u, with about 7890 non-H atoms in the asymmetric unit and 1.65Å of data resolution), and also to succeed with 1buu, a protein with 1283 non–H atom in the asymmetric unit and 1.92Å data resolution”       

Comment 2: I found most of the sentences incomplete and grammatically incorrect. Please rewrite and check for the same in the manuscript text body. In the sentence “To face this important problem, automated pipelines exist for discovering and preparing a large number of search models”, please exemplify certain examples for the model.   

Comment 3: Authors could not described clear objective in introduction section. The content is confusing and inconsistence. The sentences “This is the fourth paper of a series dedicated to the automatic crystal structure solution of macromolecular structures. Why did we focus our recent activity on this subject? Certainly not for encouraging users to no longer worry about methodologies” must be written to addressee the objectives of the study. Moreover, please describe the advancement and advantages of the current program over previous reports.  

Comment 4: The sentence “The reader, however, should not assume that REMO22 works without any directive: these may be used for defining or modifying the estimated number of model copies in the target asu, for varying figures of merit, for exploring the rotation and the translation space by using different protocols, for modifying the MR model, etc” is too long and unclear to readers. In the sentence “because it frees up resources for tackling other tasks.” The “other tasks” must be defined.      

Comment 5: In introduction, authors mentioned “Papers I-III”. These papers must be cited if published before. I recommend to write shortly about these papers. There is no connectivity between the reported papers and the current finding. Explain these in details. The sentence “recently published structures” must be cited.      

Comment 6: In the manuscript text body, there are several abbreviations used and these must be expanded when appeared first in the manuscript.     

Comment 7: In table 2. The values of “overall” is slightly different from real figure. Please recheck to avoid any error. Figure 1 is poorly presented. I suggest to keep uniformity in position.  

Comment 8: The caption of table 4 should be shortened. It’s too long. I suggest to write few information as footnote. Figure 2 caption must be expanded. In figure 2, bond line and dimensional are missing. I suggest to replace figure 2 with more information labeled. Similarly, figure 3 needs to be revised.      

Comment 9: For the sentence “According to Matthews, survey on 116 different proteins, the fraction of crystal” I did not find any citation?

Comment 10: Authors focused to run the program for nucleic acids including DNA and RNA. Could you please explain accuracy and sensitivity of the model for specific DNA or RNA? How did you validate your findings for its accuracy and sensitivity in data generated?

Author Response

(The authors gave the same response as above.)

Round 2

Reviewer 2 Report

Authors have considered comments and suggestions. The revised version is suitable for publication